# Focus image scanning microscopy for sharp and gentle super-resolved microscopy

Giorgio Tortarolo[1,5,6], Alessandro Zunino [1,6], Francesco Fersini [1,2], Marco Castello[1,3], Simonluca Piazza[1,3], Colin J. R. Sheppard [3], Paolo Bianchini [3], Alberto Diaspro [3,4], Sami Koho[1] & Giuseppe Vicidomini [1] ✉

To date, the feasibility of super-resolution microscopy for imaging live and thick samples is still limited. Stimulated emission depletion (STED) microscopy requires high-intensity illumination to achieve sub-diffraction resolution, potentially introducing photodamage to live specimens. Moreover, the out-of-focus background may degrade the signal stemming from the focal plane. Here, we propose a new method to mitigate these limitations without drawbacks. First, we enhance a STED microscope with a detector array, enabling image scanning microscopy (ISM). Therefore, we implement STED-ISM, a method that exploits the working principle of ISM to reduce the depletion intensity and achieve a target resolution. Later, we develop Focus-ISM, a strategy to improve the optical sectioning and remove the background of any ISM-based imaging technique, with or without a STED beam. The proposed approach requires minimal architectural changes to a conventional microscope but provides substantial advantages for live and thick sample imaging.

The vast family of fluorescence optical microscopy techniques stands as an invaluable tool for addressing various biological questions, enabling the dynamical observation of bio-molecular processes in living cells—on their own or as part of a whole organism. Within this family, super-resolution (SR) microscopy techniques have opened up new exciting perspectives by overcoming the classical diffraction limit of spatial resolution—about half the wavelength of the fluorescence light[1,2]. Confocal laser scanning-microscopy (CLSM)[3] was one of the first SR methods—at least theoretically. By illuminating the sample point-by-point and collecting the fluorescence light through a small pinhole, the spatial resolution could be improved by up to a factor of two below the classical diffraction limit—as for structured illumination microscopy (SIM). However, a closed pinhole leads to a strong reduction in signal-to-noise ratio (SNR), effectively precluding the resolution enhancement in a real-life scenario. Indeed, the historical success of CLSM is mostly due to its optical-sectioning capability, rather than for the resolution improvements it theoretically provides.

The long standing trade-off between spatial resolution and SNR has been recently overcome by image scanning microscopy (ISM). Such revolution has been made possible by substituting the typical single-element detector with a detector array, for which each element acts as an individual pinhole. In this case, no fluorescence signal is lost—since each detector element contributes to the collected signal—and the small size of the individual detector elements guarantees the resolution enhancement. The detector array collects a small wide-field image (i.e., a micro-image) of the illuminated region, justifying the name of the method as ISM . The additional information provided by the detector enables the reconstruction of the sample's image with a resolution enhancement close to the two-fold theoretical limit of CLSM, without sacrificing SNR. The method used for the reconstruction is known as pixel-reassignment (PR). Theoretically introduced by Sheppard in the 80s[4], and experimentally demonstrated by Enderlein in 2010[5] with a conventional camera (~kHz frame-rate), ISM became mainstream only with the introduction of fast (~MHz frame-rate)

[1]Molecular Microscopy and Spectroscopy, Istituto Italiano di Tecnologia, Genoa, Italy. [2]DIBRIS, University of Genoa, Genoa, Italy. [3]Nanoscopy & NIC@IIT, Istituto Italiano di Tecnologia, Genoa, Italy. [4]DIFI, University of Genoa, Genoa, Italy. [5]Present address: Laboratory of Experimental Biophysics, EPFL, Lausanne, Switzerland. [6]These authors contributed equally: Giorgio Tortarolo, Alessandro Zunino. ✉e-mail: giuseppe.vicidomini@iit.it

detector arrays, such as the AiryScan[6] and single-photon avalanche diode (SPAD) array detectors[7,8]: these effective super-resolved ISM implementations showed compatibility with multi-species, three-dimensional, and two-photon excitation imaging[9,10]. At the same time, the SNR enhancement allows reduction of the excitation beam intensity, improving compatibility with live-cell imaging[11,12]. Furthermore, detector arrays allow the combination of ISM with fluorescence life-time imaging to increase further the information content[11,13]. Despite its benefits, the maximum resolution of ISM is still bounded. A class of approaches which can surpass the two-fold resolution enhancement of ISM is image deconvolution. Notably, in the 80s, Bertero[14] proposed a deconvolution method which works on the micro-images to achieve twice the resolution power of a conventional laser-scanning microscope.

An entirely different class of SR microscopy concepts (referred to as diffraction-unlimited microscopy or nanoscopy) have not only surpassed the classical resolution limit, but also enabled theoretically unlimited resolution[15]. Stimulated emission depletion (STED) microscopy[16,17] is one of the most versatile of such nanoscopy techniques, since it can attain multi-species and fast imaging with tunable resolution—from the classical diffraction-limit down to a theoretical molecular resolution. A STED microscope shares similar laser-scanning architecture to CLSM, but the Gaussian excitation beam is co-aligned with a second vortex beam—known as the STED beam—and both beams are scanned across the sample. Because the STED beam induces stimulated emission and depletes the fluorescence signal from the peripheral region of the excitation spot, the effective fluorescent region reduces in size well below the diffraction limit. Notably, the higher the STED beam intensity, the smaller the effective fluorescent region, and ultimately the better the spatial resolution. However, high STED beam intensity may cause photo-bleaching and ultimately photo-toxicity to the samples. Additionally, thick samples may emit out-of-focus fluorescence background decreasing the contrast of faint in-focus fluorescence signals. The contrast further reduces if the STED beam excites directly the specimen, thus generating additional anti-Stokes background, namely the fluorescence generated by the depletion beam[18]. In other words, increasing the optical resolution comes at the cost of photo-damaging the sample and reducing the signal-to-background ratio (SBR), hence hindering the feasibility for long-term STED imaging in living cells and STED imaging in thick samples, respectively. Several STED microscopy implementations have been proposed to mitigate the photo-damage and the SBR reduction[19]: time-resolved[20–25] and subtraction methods[26,27] remove the incomplete depletion background, aiming to reduce the STED beam intensity necessary to achieve a target resolution. Tomographic STED microscopy[28] obtains comparable intensity reductions by fusing multiple STED images collected with efficient two-dimensional STED beam intensity distributions. Synchronous detection schemes[22,23,29,30] remove the anti-Stokes background. Adaptive smart scanning schemes[31–33] decrease the overall specimen illumination. Tailored three-dimensional STED beam intensity distributions[34,35] chop the out-of-focus background. Among these discussed techniques, only time-resolved STED microscopy has now become a gold standard, being implemented in all commercial systems. The reasons behind this success lie in the simplicity of the technique: it requires only to register the temporal dynamics of the fluorescent signal, adding an additional temporal dimension to the image dataset. The other approaches provide great benefits but require a significant increase in technical complexity, leaving the scientific community still in need of easy strategies to improve the compatibility of STED microscopy with live-cell and thick samples. Recently, new diffraction-unlimited ISM-based techniques have been proposed, either exploiting photon-coincidences[36] or fluorescence fluctuations[37]. These techniques do not require laser powers as large as those of STED microscopy. However, they require a long pixel-dwell time (≥ms) to achieve high SNR,

while resolution enhancement greater than 2.5 has not been demonstrated, thus limiting their practical applications in the current development state. In this work we combine ISM with STED microscopy (Fig. 1a) to achieve two significant benefits: (i) the reduction of the STED beam intensity necessary to achieve a resolution target, and (ii) the suppression of out-of-focus background. Importantly, we note and demonstrate that the background rejection strategy introduced here can be also applied to ISM imaging alone—in other words, not combined with STED microscopy. Our architecture is based on a fast SPAD array[7] that enables the acquisition of a small wide-field image of the excited region, one per each scan point (Fig. 1b). These small images, which we call micro-images, in general contain information on the lateral and axial structure of the sample, enabling the reconstruction of an image with enhanced content. In detail, (i) we applied the adaptive pixel-reassignment (APR) method to improve the spatial resolution while keeping a relatively low STED beam intensity, thus potentially reducing photo-damage; (ii) we introduced a classification method to separate the out-of-focus background from the in-focus signal, thus improving optical-sectioning. More generally, we investigated in detail the multidimensional ISM dataset from different points of views. The 25 scanned images encode the information of the shift vectors, a key element for APR reconstruction (Fig. 1c), as described in the results sections. We prove that their role is even more crucial for STED-ISM, being very sensitive to the depletion power and thus hindering different approaches of pixel reassignment. Here, we demonstrate that, after a successful reassignment, the micro-images directly encode the sample's axial position for any STED intensity. We exploited this critical insight to develop focus-ISM, a two-step algorithm. First, it implements the well-established APR method and then a classification algorithm to remove the out-of-focus light, while leaving intact the in-focus signal (Fig. 1d). We tested the focus-ISM method on calibration samples and fixed/living cells, achieving sharper STED microscopy images at any STED beam intensity. While the benefits of STED-ISM are reduced in the case of a high STED beam intensity, the focus-STED-ISM method improves optical sectioning at any STED beam intensity—also in the limiting case of the simple confocal modality. Lastly, we further improved the contrast and the quality of the reconstructed images by means of a multi-image deconvolution algorithm, incorporating the out-of-focus background obtained thanks to focus-ISM as prior information.

## Results

We obtained the results presented in the following sections using a custom STED microscopy setup incorporating an asynchronous readout SPAD array detector (Supplementary Fig. 1). Before commenting the findings of our work, we briefly review the state-of-the-art of ISM with a SPAD array. In detail, the detector array records a bi-dimensional micro-image $i(\mathbf{x}_d|\mathbf{x}_s)$, with $\mathbf{x}_d = (x_d, y_d)$ the detector space, for each point of the scanning space $\mathbf{x}_s = (x_s, y_s)$, effectively adding two additional spatial dimensions to the conventional image dataset. Thus, the ISM multidimensional dataset can be regarded both as a collection of micro-images $i(\mathbf{x}_d|\mathbf{x}_s)$—one for each scan point $\mathbf{x}_s$—or as a collection of scanned images $i(\mathbf{x}_s|\mathbf{x}_d)$—one for each detector element at position $\mathbf{x}_d$. The former perspective is the one adopted to apply the PR approach in the opto-mechanical ISM implementations[38] and, more generally, in the original PR idea (see Supplementary Section 1). In ISM, the photons collected by each pixel of the detector are shifted from their most likely origin $\mathbf{x} = \mathbf{x}_s - \mathbf{x}_d + \boldsymbol{\mu}'(\mathbf{x}_d)$. Assuming no Stokes shift and Gaussian point spread functions (PSFs), namely the distribution of light on the sample (excitation PSF) and on the detector (detection PSF), the so-called micro-image shift-vector $\boldsymbol{\mu}'(\mathbf{x}_d)$ equals $\mathbf{x}_d/2$ and PR can be implementing by demagnifying twice the micro-image. More recently, we introduced the concept of APR to generalize PR to any imaging conditions[11,39]. The APR approach regards the ISM dataset from the scanned images point of view $i(\mathbf{x}_s|\mathbf{x}_d)$ which is key for

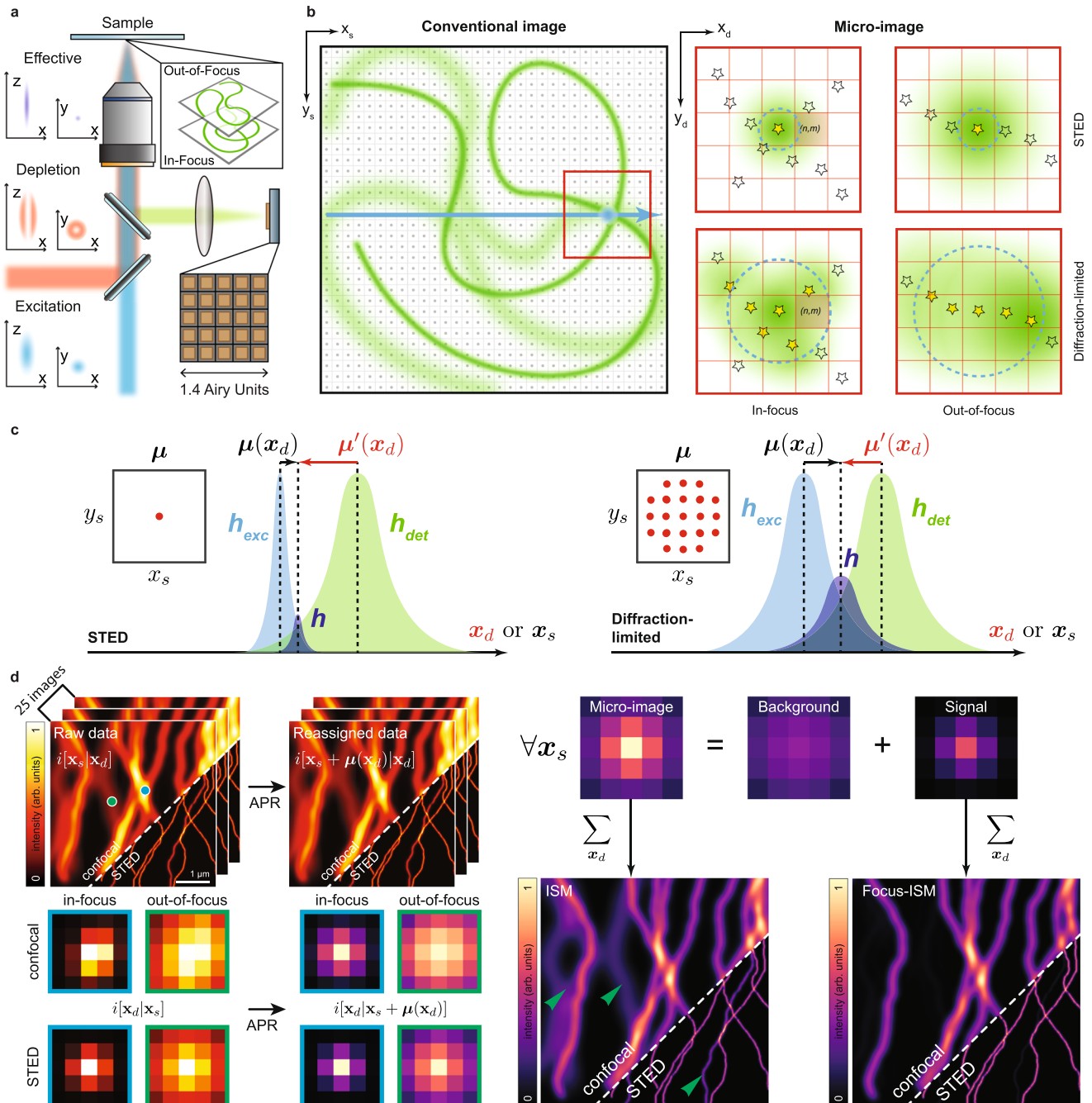

**Fig. 1 | Focus-ISM imaging.** In (**a**), we present a sketch of the STED-ISM microscope, equipped with a 5 × 5 detector array. In (**b**), we present a sketch of the image formation process in our setup. The sample—represented with an in-focus (sharp) and an out-of-focus (blurred) component—is raster scanned with the excitation beam. Each scan point (dark square) is related to a micro-image of the illuminated region (red square). The blue circle and the green halo represent, respectively, the excitation PSF and the emission of the excited fluorophores (yellow stars). In (**c**), we depict the (adaptive) pixel-reassignment concept for a specific detector element $(n, m)$. The shift vector $\boldsymbol{\mu}$ of a scanned image is the maximum position of the product of excitation and detection PSF. The higher is the STED power, the smaller is the excitation PSF, the shorter is the shift vector. We also show the micro-image shift vector $\boldsymbol{\mu}'$. In (**d**), we present the steps of focus-ISM reconstruction. The APR algorithm registers the scanned images of the ISM dataset. APR affects also the micro-images, encoding uniquely the axial information of the sample. The Focus-ISM algorithm classifies the photons of each post-APR micro-image either as background or signal. Summing the pixels of the signal micro-images generates the Focus-ISM image. The green arrows indicate the out-of-focus filaments, no longer present in the focus-ISM image.

understanding the ISM image formation. The images $i(\mathbf{x}_s|\mathbf{x}_d)$ are mutually shifted by the quantities known as the scanned-image shift vectors $\boldsymbol{\mu}(\mathbf{x}_d)$. In short, the APR method registers the scanned images and generates the ISM result, with enhanced resolution and SNR. The APR algorithm calculates the shift vectors from the data by finding the shift that maximizes the similarity with a reference image, in our case the one generated by the central element of the detector array. For this

reason, the APR method adaptively finds the best estimates of the shift vectors directly from the images, taking inherently into account non-idealities—such as the Stokes shift or optical aberrations—without any theoretical assumption. This aspect is of paramount importance to combine ISM with STED microscopy, because the value of the shift vectors strongly depends on the STED power, making digital ISM the only technique compatible with STED microscopy.

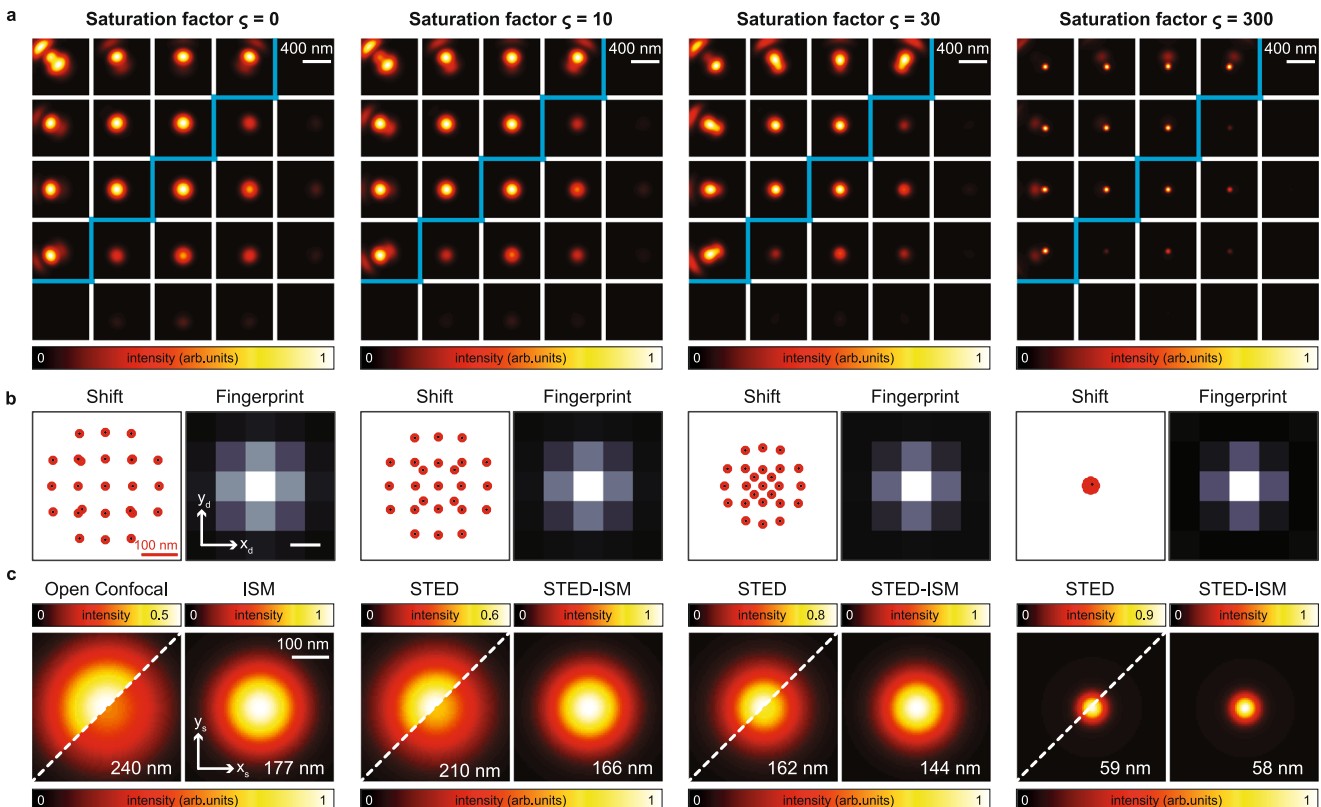

**Fig. 2 | STED-ISM principle.** In (**a**), we show simulated images of the PSF $h(\mathbf{x}_s|\mathbf{x}_d)$ for each detector element. In (**b**), we show the shift vectors, calculated using the adaptive pixel reassignment algorithm, and the fingerprint, calculated by summing all the $\mathbf{x}_s$ points of the above images. In **c**, we compare the PSF obtained with a single-element detector (open pinhole configuration) with the PSF reconstructed with the ISM method. The upper diagonal is normalized to itself, and the lower diagonal is normalized to the maximum of the corresponding ISM reconstruction. The numerical value is the FWHM resolution of the PSF. Each result is shown for increasing saturation factor ($\varsigma$), from left to right.

## Reducing intensity with adaptive pixel-reassignment ISM

In this section, we demonstrate how the concept of APR can be beneficial to STED microscopy, enabling a target resolution at lower STED beam intensity. To this end, we simulated at increasing depletion power the in-focus scanned PSFs (Fig. 2a), the relative shift vectors (Fig. 2b), and the PSFs of the conventional STED and of the STED-ISM image, reconstructed with the APR method (Fig. 2c). We obtained the conventional STED images by summing all the scanned images, thus discarding the micro-image information, as it would happen with a single-element detector. The shift vectors strongly depend on the STED beam intensity: the higher the depletion power (equivalently, the saturation factor), the smaller the effective fluorescent spot, and the shorter the shift vectors. Based on the APR concept, the shift vectors reflect the maximum position of the PSFs of each scanned image. Since the excitation PSF shrinks down to a single point for increasing STED beam intensity, its product with any detection PSF shrinks as well and its maximum position approaches the optical axis. In other words, in the case of high STED beam intensities, all scanned PSFs depend mainly on the effective excitation PSF and the influence of the detection PSF becomes negligible. Thus, the scanned images vary in SNR but no longer in position. This insight shows that the adaptive pixel reassignment operation is beneficial for STED microscopy mainly for a specific range of STED beam intensities. More precisely, the spatial resolution and SNR of the STED-ISM reconstruction are improved with respect to the conventional STED counterparts for mild STED beam intensities (Fig. 2c, Supplementary Fig. 2). High STED beam intensities lead to very short shift vectors and ultimately to negligible benefits from the APR method.

We also calculated the so-called fingerprint at increasing STED beam intensities (Fig. 2b). We have defined the fingerprint of an ISM dataset as the sum of all micro-images[11] and it measures the convolution of excitation and detection PSF. Thus, it describes the distribution of photons on the detector array with no influence from the specimen −except for an intensity scale factor. We calculated the fingerprints by integrating along the scanning dimensions ($x_s$, $y_s$) the ISM dataset of the simulated point-source sample. As expected, increasing the STED power reduces the width of fingerprint, reflecting the shrinking of the effective excitation PSF. Eventually, for extreme STED beam intensity, the fingerprint identifies with the detection PSF. To validate our STED-ISM approach, we continued by acquiring super-resolved images of various samples. We imaged 20 nm diameter fluorescent beads with increasing STED beam power (Fig. 3a, b). Consistent with the simulations, for relatively low STED beam power, the STED-ISM image shows better SNR and better resolution, when compared to the conventional STED counterpart (obtained by summing all 25 channels). For high STED beam powers the benefits of STED-ISM vanish, as quantitatively confirmed by the resolution and signal gain graphs reported in Fig. 3c (see also Supplementary Fig. 2). We confirmed these results by performing STED-ISM measurements of a sample of fixed Hela cells (Fig. 3d). It is important to note here that the successful STED-ISM reconstruction relies heavily on our APR method, a blind and parameter-free image phase-correlation algorithm[11,39], able to retrieve the shift vectors directly from the scanned images. In general, the adaptive approach compensates for possible misalignments and other non-idealities of the optical system. Notably, the experimental shift vectors are drastically different from the ideal ones of Fig. 2b, highlighting the crucial role of an adaptive reassignment[11]. As discussed above, in the context of STED-ISM the APR method also implicitly accommodates the strong dependency of the shift vectors on the STED beam powers. In contrast, (i) mechanical ISM implementations

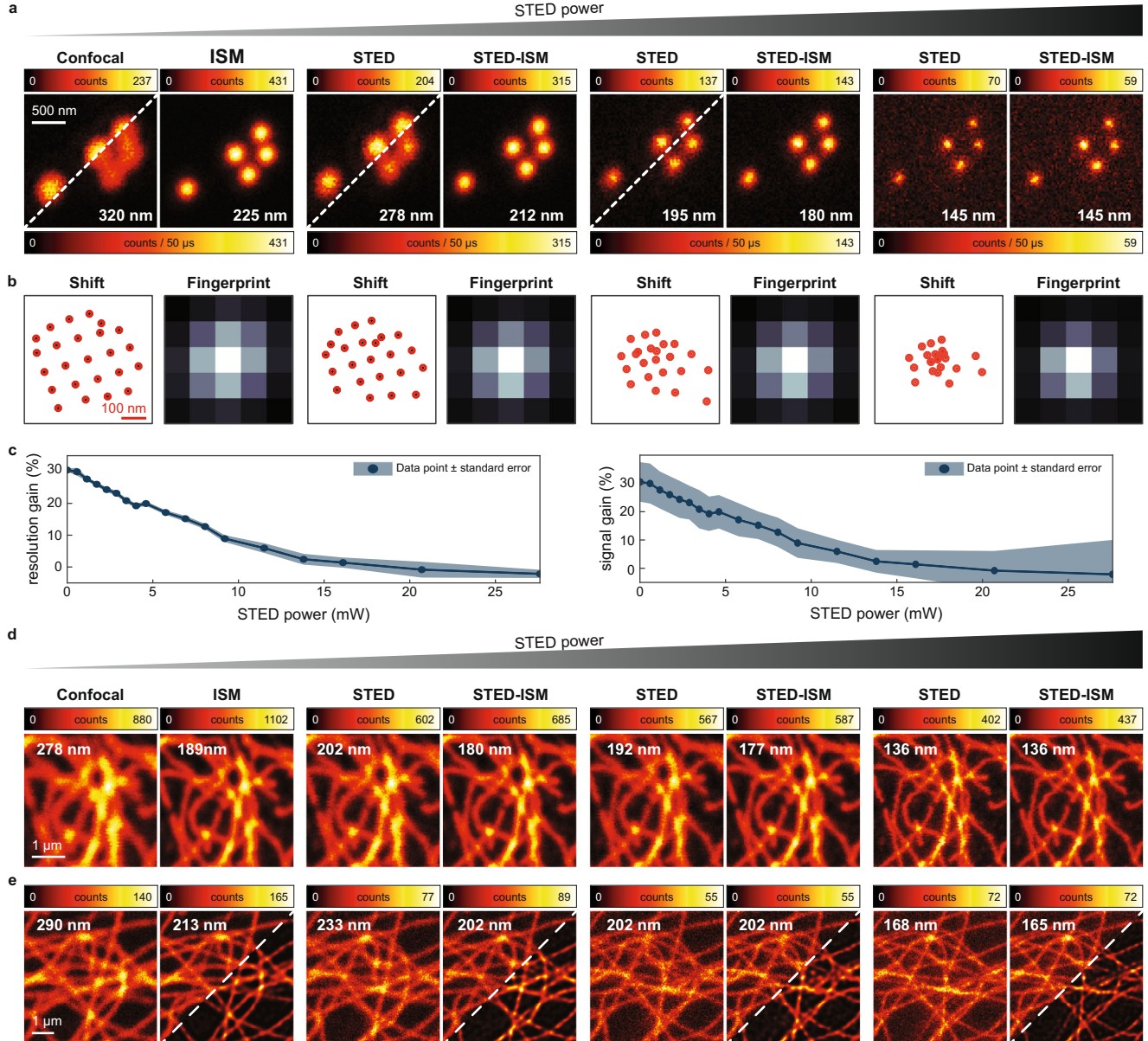

**Fig. 3 | STED-ISM imaging.** In (**a**), we compare raw images of fluorescent beads with the corresponding ISM reconstructions. In (**b**), we show the shift vectors and fingerprint calculated from an image of the beads. In (**c**), we show the resolution gain (left) and the signal gain (right) of STED-ISM with respect of STED, for increasing STED powers. We measure the resolution and the signal, respectively, as the FWHM and the peak value of the successful fit of the fluorescent bead to a Gaussian curve. The graphs report the average values of multiple beads with the corresponding standard errors. In (**d**), we show detailed regions of images of tubulin-labeled fixed cells. In (**e**), we show detailed images of living Hela cells with SIR tubulin labeling. More specifically, we compare raw STED (left), STED-ISM (right, upper corner) and the result of multi-image deconvolution STED-ISM⁺ (right, bottom corner). All results are shown with increasing STED power, from left to right. The full images are shown in Supplementary Figs. 3, 4. The values in white are the images' resolution, estimated using a fit to a Gaussian model (**a**) or Fourier ring correlation (**d** and **e**).

should rely on proper modelling or prior calibration of the STED microscope's effective PSF, as a function of the STED beam intensity and of the general experimental conditions; (ii) optical ISM implementations would require a change of the demagnification factor, which is impractical to achieve.

To investigate the concrete advantages of STED-ISM over conventional STED microscopy, we explored the case when one is concerned most about the number of stimulating photons delivered to the sample: live-cell imaging (Fig. 3d, Supplementary Fig. 4). Also in this case, we report an enhancement of SNR and spatial resolution of the resulting STED-ISM images with respect to raw STED counterparts. The results are further improved by applying our multi-image deconvolution algorithm[11,40], here completely parameter-free thanks to the PSFs estimation via Fourier ring correlation (FRC) analysis[41]. Moreover, we

were able to perform extended STED-ISM time lapses of live Hela cells without inducing any noticeable photo-bleaching effect, given the reduced STED beam intensity necessary to obtain the target resolution (Supplementary Fig. 5)

## Removing background with focus-ISM

In this section, we regard at the ISM dataset from the perspective of the micro-images $i(\mathbf{x}_d|\mathbf{x}_s)$ in order to effectively improve the optical sectioning of ISM. Our method, named focus-ISM (f-ISM), removes the out-of-focus background analyzing each micro-image. First, we show the working principle and the feasibility of f-ISM considering the case of ideal STED, namely the case of point-like excitation (or, equivalently, infinite depletion power). Later, we generalize our method to STED microscopy at any depletion power and to conventional ISM. As a

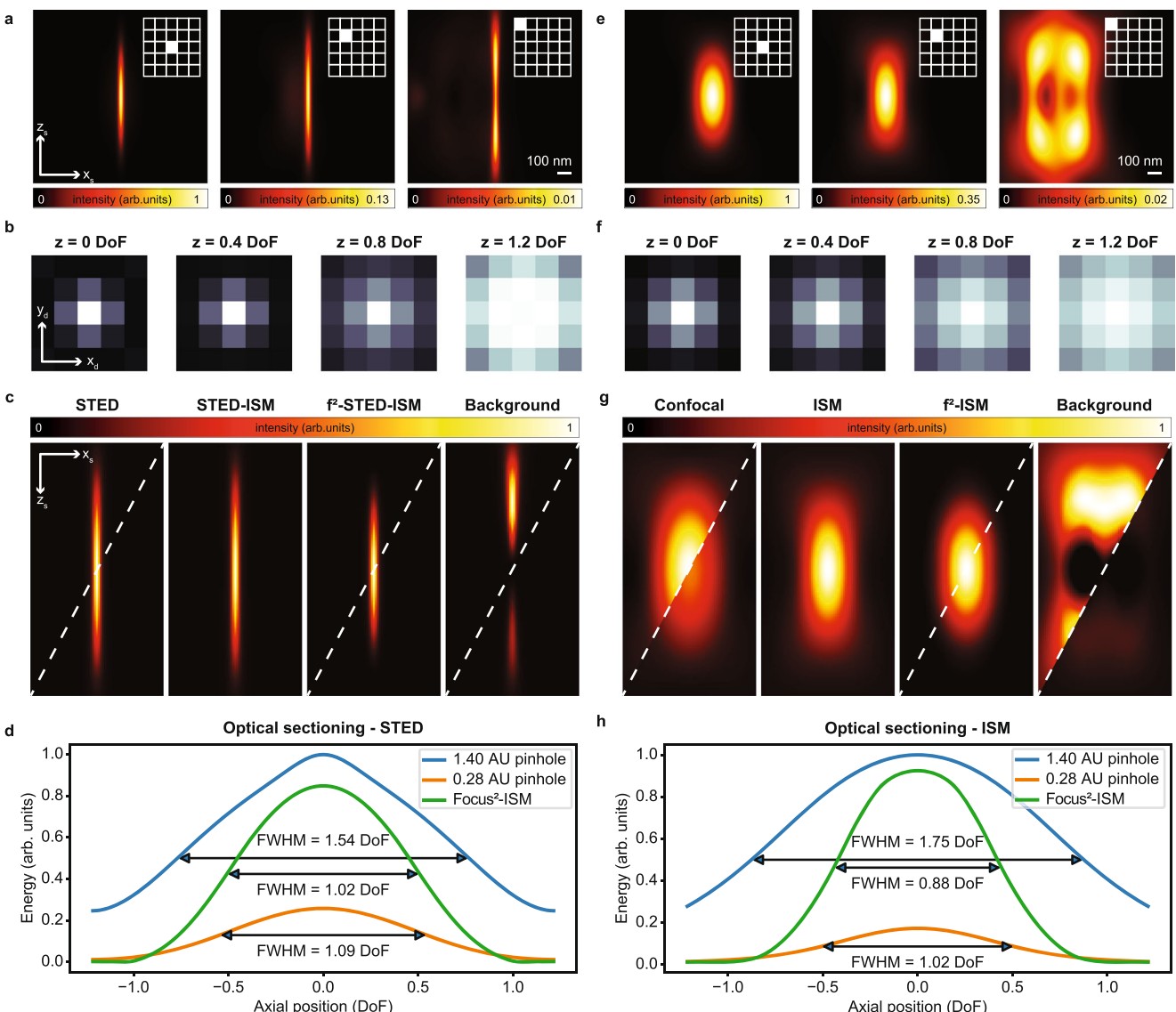

**Fig. 4 | Focus-ISM principle.** In (**a**), we show STED PSFs ($\varsigma = 300$) related to the detector element highlighted with the white box. The $z, r$ images are obtained by sectioning the 3D PSFs along the main diagonal. In (**b**), we show the fingerprint at different axial positions, calculated by summing all the scan points from the corresponding sections of the STED PSF. In **c**, we show the reconstructed STED PSFs. From left to right: raw STED (open pinhole), STED-ISM, focus-ISM, and removed background. In (**d**), we show the energy of each axial plane of the simulated STED PSFs, obtained by integrating the volumes along the $x_s$ and $y_s$ dimensions. In (**e**), we show the (normalized) confocal PSFs corresponding to the detector element highlighted with the white box. In (**f**), we show the fingerprint at different axial positions, calculated by summing all the scan points from the corresponding sections of the confocal PSF. In (**g**), we show the reconstructed confocal PSFs. From left to right: raw confocal (open pinhole), ISM, focus-ISM, and removed background. In (**h**), we show the energy of each axial plane of the simulated ISM PSFs, obtained by integrating the volumes along the $x_s$ and $y_s$ dimensions. Note that the ISM and STED-ISM curves are not explicitly reported, being identical to the open pinhole curve (1.40 A.U.). Indeed, pixel reassignment does not affect the total photon counts. The top-left corners of the images of the PSFs are normalized to themselves, while the bottom-right corners are normalized to the ISM reconstruction.

matter of fact, our method is well suited also to any ISM-based technique.

The core idea behind focus-ISM consists in observing the distribution of the light on the detector array to distinguish the axial position of the emitters. We simulate the three-dimensional (3D) scanned PSFs of an ideal STED microscope to demonstrate the working principle (Fig. 4a). The central element of the detector array mostly contains in-focus signal, while the out-of-focus light dominates the outer elements. We can extract the same information by calculating the axial fingerprints (Fig. 4b), namely, by integrating, at different depths $z_s$, the 3D scanned PSFs over the scanning coordinates ($x_s, y_s$). When the emitter is in focus, the central pixels contain most of the signal. The farther the distance from the focal plane, the more the

outer pixels of the fingerprint are populated with photons. We can quantify this trend by calculating the ratio of the intensity of the outer pixels to the intensity collected by the central pixel (Supplementary Fig. 6a). As expected, the outer pixels collect more light as depth increases. Interestingly, the micro-images $i(\mathbf{x}_d|\mathbf{x}_s)$ of each scan point $\mathbf{x}_s$ show similar behaviour (see Supplementary Section 2, Supplementary Fig. 7a): in the case of ideal STED, both the micro-images and the fingerprint coincide with the detection PSF, weighted by the sample brightness. Indeed, if the effective excitation spot (the excitation PSF) is small enough to ideally excite only a single point, the corresponding wide-field micro-image is the detection PSF of the microscope, centred at the scan point. Similarly, the fingerprint is the sum over the scan of all the micro-images, and contains the same information with a higher

SNR. For ideal STED microscopy, lateral confinement of the fluorescent region also occurs outside the focal plane (Supplementary Fig. 8). Thus, the out-of-focus micro-images and fingerprints also correspond to the out-of-focus detection PSF. These observations suggest that the information about the axial position is encoded in the lateral distribution of the light at the detector plane, and can be exploited pixel-by-pixel to remove the out-of-focus background. Indeed, each micro-image can be seen as the linear combination of a narrow in-focus component and a broad out-of-focus component.

In the following, we present a naive approach to exploit the relation between the distribution of the signal on different detector elements and its origin on the optical axis. Later, we discuss a more precise approach to identify and remove the out-of-focus background from each micro-image, and consequently to the reconstructed STED-ISM image.

The out-of-focus light is distributed broadly across the detector elements, and outer elements do not register in-focus light. This observation suggests a simple way to estimate the background—namely using a flat out-of-focus micro-image. Our first algorithm ($f^1$-ISM) estimates the background by calculating the average of the signal collected by the outer elements of the detector array. The background value is then subtracted from the inner elements. In detail, we use all the outer-ring elements to extract the average background per scanpoint. The simplicity behind this strategy, which only requires a few basic arithmetic operations, enables a fast estimation of the background, paving the way for real-time focus-ISM. However, the approximation behind this approach is crude and might lead to non-physical results, such as pixels with negative photon counts.

Our second approach is more sophisticated and consists in modelling the axial STED fingerprint (i.e., the detection PSF) with a two-dimensional Gaussian distribution, which is known to be a robust approximation[42] in the absence of strong aberrations. We tested the Gaussian model by fitting each simulated axial fingerprint (Supplementary Fig. 6b). As expected, the standard deviation increases for increasing depth. Still, it is approximately constant around the focal plane for roughly one depth-of-field thanks to the 3D structure of the depletion beam (Supplementary Fig. 8). Our second algorithm ($f^2$-ISM) fits each micro-image to the weighted sum of two Gaussian functions, the first with a broad standard deviation—used to model the out-of-focus component—and the second with a narrow standard deviation—able to model the in-focus light. The standard deviation of the in-focus distribution is kept fixed and precalibrated either theoretically or experimentally (Supplementary Fig. 9). The standard deviation of the out-of-focus distribution is typically a free fitting parameter, but it can also be kept fixed with a user-selectable value (Supplementary Fig. 10). We obtain the final $f^2$-ISM image by integrating only the portion of the signal fitted to the in-focus term. This second approach is computationally more expensive but enables a more precise classification of the in/out-of-focus components. Additionally, we applied physically meaningful constraints (such as conservation of the photon flux) to guarantee the non-negativity for the pixels of the reconstructed image. Because for ideal STED the shift vectors are null, in this case we did not apply adaptive pixel reassignment before applying focus-ISM.

To demonstrate the capabilities of our algorithm, we applied the $f^2$-ISM method to the simulated 3D scanned PSF of an ideal STED microscope (Fig. 4c). We recall that in the case of ideal STED the APR method is not necessary. Notably, compared to the STED-ISM PSF, the loss of photons from the focal region is almost negligible, but the out-of-focus light is strongly suppressed, effectively enhancing the optical sectioning of the microscope without reducing the signal. This phenomenon can be quantified by analyzing the curves of the PSFs integrated respect to the lateral dimensions ($x_s, y_s$), shown in Fig. 4d. Indeed, the width and the height of such curves indicates, respectively, the background rejection and the signal preservation of the corresponding imaging technique. Notably, Focus-STED-ISM suppresses the

out-of-focus light better than STED with a closed pinhole. At the same time, most of the in-focus signal is preserved, while STED with a closed pinhole also rejects a great fraction of the in-focus light. The reliability of our result can also be appreciated by simulating the imaging of a three-dimensional filament network (Supplementary Fig. 11a).

We confirmed the benefit of focus-ISM by performing STED imaging on a sample of tubulin-labeled fixed Hela cells (Fig. 5). This sample is of particular interest because tubulin filaments wrap in three dimensions around the nucleus. In this case, the out-of-focus background significantly degrades the STED-ISM image. The focus-ISM method recovers the contrast significantly by estimating and removing the out-of-focus background. As anticipated, $f^1$-STED-ISM may introduce negative intensity values, while this problem vanishes for $f^2$-ISM. Furthermore, $f^2$-ISM provides higher signal-to-noise ratio (SNR) than $f^1$-ISM, as confirmed by the higher peak counts in the image. We further improve image quality with deconvolution: we adapted a multi-image deconvolution algorithm to introduce the background as an a priori information ($f^+$-ISM), which can be estimated with any of the discussed approaches. Notably, using image deconvolution, negative values do not appear, even if the background is estimated using the $f^1$-ISM method.

The advantages of focus-ISM in terms of optical sectioning are even more clear when performing three-dimensional imaging. We show the result of volumetric STED imaging of a tubulin-labeled fixed Hela cell (Fig. 6a, Supplementary Fig. 12). The reconstructed volume, obtained with the $f^2$-ISM method, presents sharper axial cross-sections and close to no out-of-focus light.

So far, we have discussed only the out-of-focus fluorescence background generated by the excitation of the sample at positions different from the focal plane. However, the depletion beam can generate non-negligible anti-Stokes fluorescence background possibly originating from any axial plane. Because of the annular-shaped distribution of the STED beam intensity, anti-Stokes fluorescence originates mainly from the periphery of the effective excitation region. Thus, the anti-Stokes background localizes in the fingerprint and micro-images similarly to the conventional out-of-focus background (Supplementary Fig. 8), and our focus-ISM approach is also able to remove the background from this source. This feature is especially beneficial in STED implementations where the anti-Stokes background is potentially dominant, such as single-wavelength two-photon excitation STED[43].

As anticipated, focus-ISM can be applied for any depletion intensity, down to the case of a standard confocal microscope. Indeed, we theoretically demonstrate that, as for ideal STED microscopy, the post-APR micro-images are equal to the fingerprint (see Supplementary Section 2) for any STED beam intensity. Thus, applying the focus-ISM classification after the APR improves the optical-sectioning at any experimental condition. An intuitive demonstration of the equivalence between post-APR micro-images and fingerprint (Supplementary Fig. 7c) follows here. Because the micro-images of an ISM dataset are just small wide-field images of the illuminated area (Supplementary Fig. 7b), their content depends on the structure and the position of the sample. Nevertheless, the result of pixel reassignment is the relocation of each pixel to the position of its emitter on the sample space $x_s$. Consequently, under the hypothesis of perfect reassignment, the pixels of the post-APR micro-images carry only the information of the same point $x_s$, as they would under point-like illumination centred at the same position. Thus, the only information content left in the post-APR micro-image is the probability of detecting a photon originating from the position $x_s$ with the detector element at position $x_d$. This probability distribution is exactly the fingerprint—apart from an intensity scale factor.

Because in the confocal case, the out-of-focus signal is distributed over a broader region than in the ideal STED case (Fig. 4d-e), the detector elements located on the outer ring do not provide a reliable

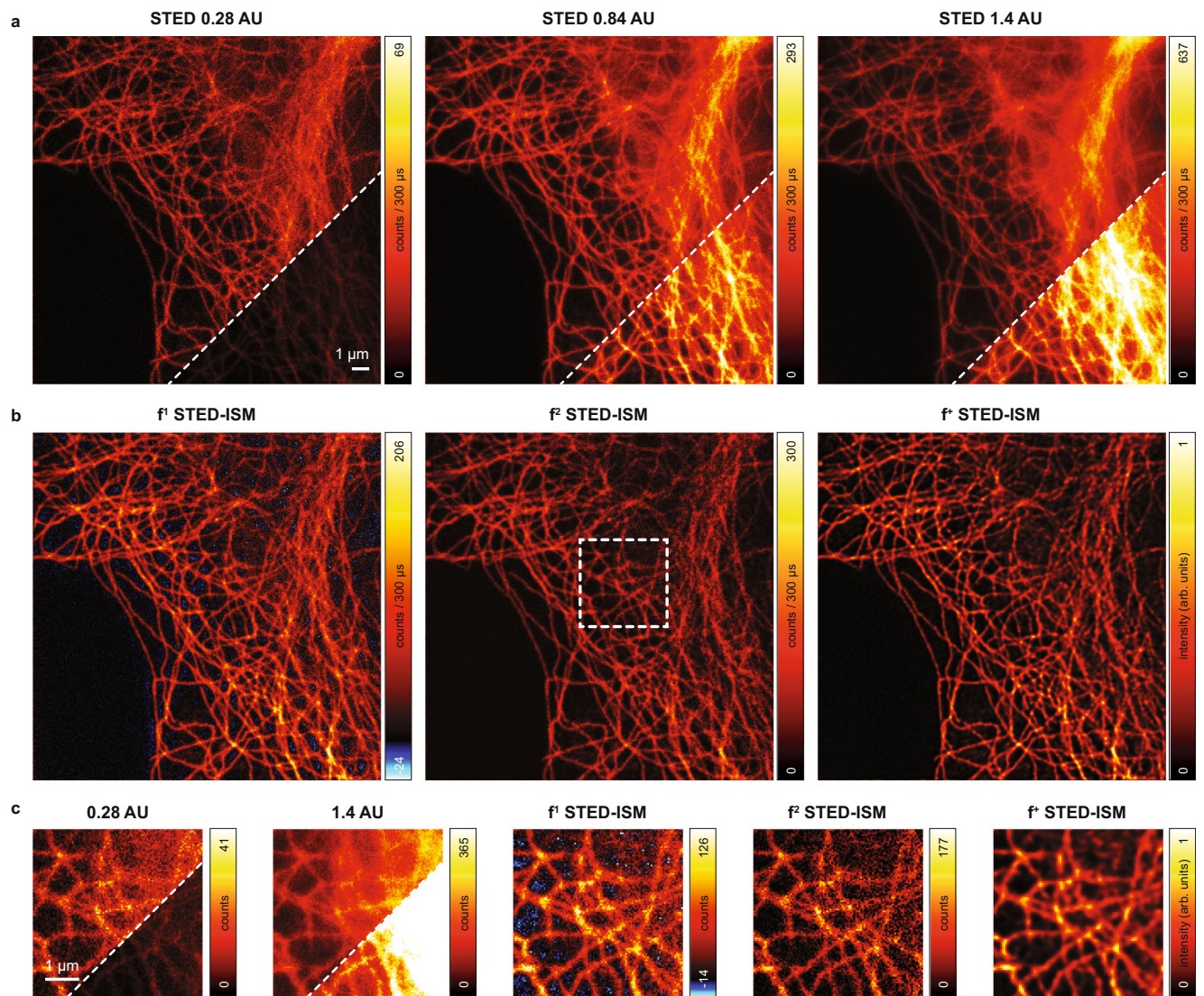

**Fig. 5 | focus-ISM for STED imaging. a** STED image of tubulin-labeled fixed Hela cell at different pinhole sizes. The bottom/right corner is normalized to the $f^2$-STED-ISM image. Even at the smallest pinhole size, the out-of-focus light hides some structure. **b** f-ISM reconstruction of the STED image. The $f^1$-ISM may introduce some negative values, here coloured blue. The $f^2$-ISM method does not produce negative values and maintains a higher photon count. The $f^+$-ISM image has been produced with a Richardson-Lucy deconvolution algorithm manually stopped at the 5th step and using the background estimated from the $f^1$-ISM method. **c** Zoomed details of the above images, corresponding to the region identified by the dashed white box.

estimate of the background, unless we use a larger detector. Thus, we analysed the confocal images exclusively with the $f^2$-ISM algorithm. As in the STED case, the confocal PSF reconstructed by focus-ISM contains the same number of photons from the focal region, but the out-of-focus light is almost completely removed (Fig. 4f). The optical sectioning curves, shown in Fig. 4h, demonstrate that also in this case Focus-ISM suppresses the out-of-focus light better than CLSM with a closed pinhole while preserving most of the in-focus signal. We also validated focus-ISM for the confocal case on synthetic images of a three-dimensional filament network (Supplementary Fig. 11b).

We also demonstrated the performance of focus-ISM in the absence of stimulated emission depletion. In detail, we performed confocal imaging of the same tubulin-labeled fixed Hela cell used for STED imaging and we show how focus-ISM removes the background efficiently without sacrificing the SNR (Fig. 6b, Supplementary Fig. 13b). A detailed quantification of the contrast improvement can be found in Supplementary Fig. 14. The analysis of the radial spectra of the images of Fig. 6 shows an enhancement of the high-frequency content over the low-frequency content upon the application of the $f^2$-ISM

algorithm. This effect is a consequence of the removal of the out-of-focus content—which being blurrier contain mostly low-frequency signal—and is the source of the contrast enhancement of our images.

Focus-ISM is drastically different from previous attempts at out-of-focus background reduction. Historically, the first effort to reduce the out-of-focus background consisted in closing a pinhole placed in front of the detector. However, as demonstrated by our results, this approach does not fully solve the problem—some background light can reach the detector even if the pinhole size is small—and compromises the SNR of the resulting image (Supplementary Fig. 15). Three-dimensional deconvolution of a confocal image is a more advanced technique and can, in some cases, be a valuable alternative to focus-ISM. In this approach, by collecting optical sections of a three-dimensional specimen, it is possible to reconstruct the volume by reassigning the photons to their origin. Despite being the most conservative solution—no signal is lost, it is just reassigned—it requires the collection of a whole three-dimensional dataset, forcing much longer acquisition times. In experiments in which it is essential to minimize the light dose (e.g., extreme resolution STED) or when time series of 2D

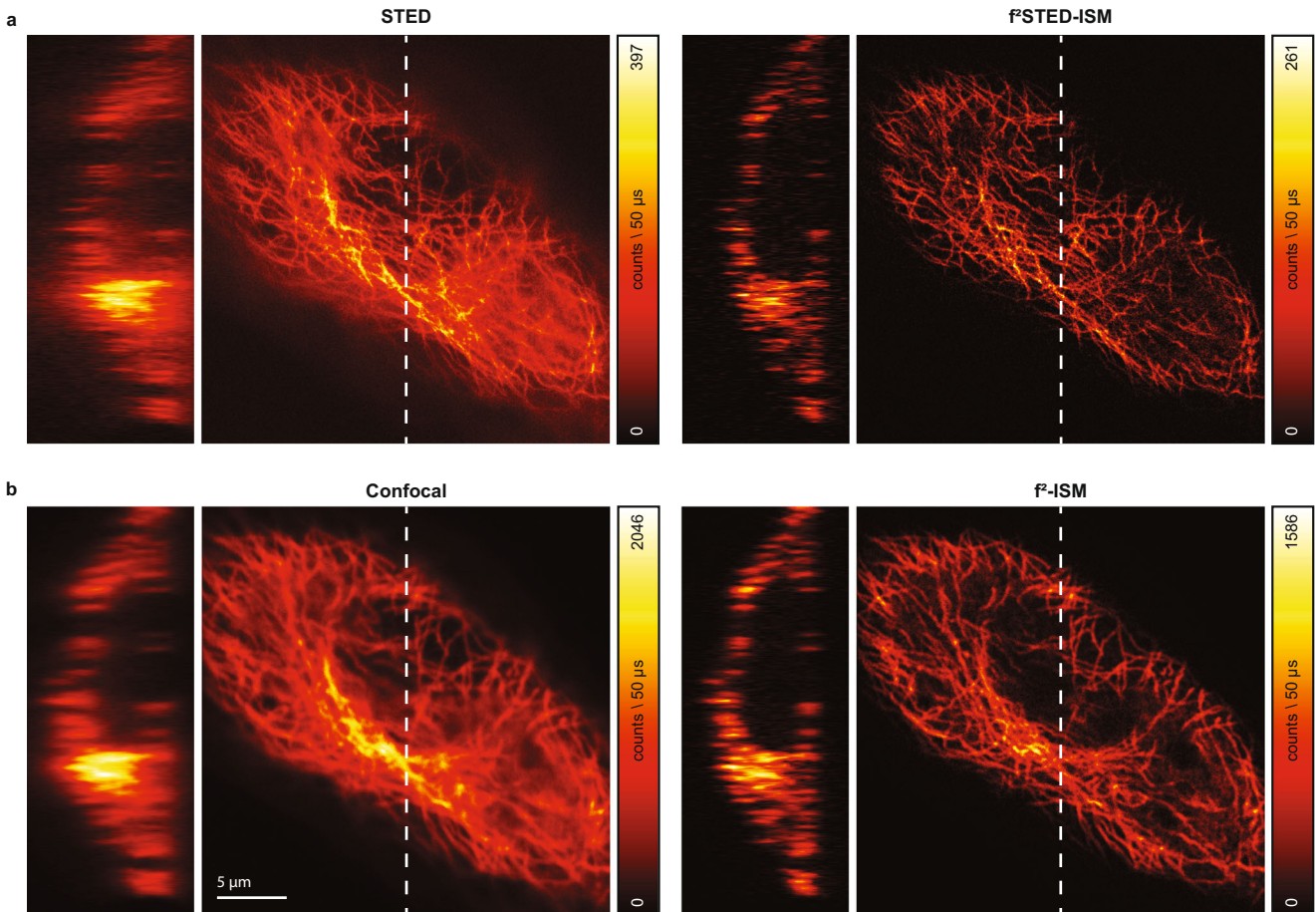

**Fig. 6 | Volumetric focus-ISM.** In (**a**), we show the STED image of a tubulin-labelled fixed Hela cell (left). The $f^2$-ISM reconstruction (right) show negligible loss of in-focus photons and greatly enhanced contrast, obtained by removing the out-of-focus content. In (**b**), we show the confocal image of the same cell (left). Even in this case, the $f^2$-ISM reconstruction (right) shows negligible loss of in-focus photons and greatly enhanced contrast. The axial cross-sections are obtained from the *x*-axis highlighted with a white dashed line.

images are the main goal, collecting a full-volume image is unfeasible, and so is 3D deconvolution. Thus, a method capable of estimating the out-of-focus background directly from a bi-dimensional image is essential. Other than focus-ISM, some other techniques have been proposed in the past. Subtraction-based methods have been circulating since the early 90s[44] and more recently revamped with the introduction of Airyscan-like detectors[45]. However, subtraction methods cannot generate spatial frequencies beyond the confocal cut-off frequency, meaning that these methods generate a contrast enhancement rather than an effective optical section. At the same time, the subtraction may introduce negative intensity values, which have no meaning from a physical point of view. The simplest escape route is to assign zero counts to the pixels with negative values, but that would break the linearity property of the image, thus precluding the possibility to associate it with a PSF. To mitigate the introduction of non-physical values, some authors scaled the subtraction term with an empirical factor[46], creating a trade-off between artefact generation and background suppression.

We fully solve these problems with the $f^2$-ISM implementation of focus-ISM, which can remove the background of a 2D image without sacrificing its SNR and with no risk of generating non-physical results. Indeed, the intensity profiles of the PSFs (Supplementary Fig. 16) confirms that the peak value in focus is negligibly affected. At the same time, the axial extension is significantly smaller. The improvement in terms of optical sectioning can be studied also in the frequency space, calculating the Modulation Transfer Functions (MTFs) shown in Supplementary Fig. 17 and Supplementary Fig. 18. The structure of the

MTFs confirms that Focus-ISM improves the optical sectioning and contrast of ISM and STED-ISM, even better than using a pinhole, without sacrifing the SNR.

Nevertheless, a critical analysis of focus-ISM is still required. Indeed, the algorithm relies on the assumption that pixel reassignment is exact. However, this condition is verified only if the scanned images are all identical but shifted and rescaled. However, in a real-case scenario, this is only approximately true. The shape of the PSFs associated with each pixel of the detector array is roughly the same for elements inside a region of about one Airy Unit. The PSF of more external elements has a distorted shape, but also carries minimal signal. Thus, the perfect reassignment hypothesis is very robust in the absence of other sources of distortions—such as optical aberrations.

## Discussion

The combination of STED microscopy with a detector array offers impressive advantages in terms of optical sectioning and minimization of the risk of inducing photo-damage, key factors for super-resolution imaging of thick and live samples.

As our results demonstrate, the boost in resolution provided by APR and STED synergically combine to enable super-resolution at lower STED beam power. Notably, this benefit is maximal at a low saturation factor. Indeed, at higher depletion power the gain in resolution is dominated by the STED effect—ideally unlimited—while the gain in resolution provided by ISM is bounded to a factor of two.

Additionally, the detector array enables the application of focus-ISM, an algorithm that is capable of discriminating the out-of-focus

light from the in-focus light. By analysing the post-APR micro-image, we are able to remove the background without introducing artefacts and while preserving the SNR. Thus, the combination of APR with focus-ISM enables high contrast and high-resolution imaging, opening unexplored scenarios for the imaging of thick samples. Remarkably, the proposed method works for any STED beam intensity−including the limiting case of an inactive STED beam. We think that our tool will be beneficial not only for ISM and STED-ISM, but also for any laser-scanning microscope potentially equipped with a detector array, such as two-photon excitation microscopes. Importantly, we believe that focus-ISM is another step towards a more refined usage of the extra-spatial information provided by the SPAD array. Indeed, in this work, we used the axial classification of the light only to remove the out-of-focus light. Nevertheless, we believe that using a more detailed mathematical model it is possible to localize precisely the position of the emitter along the optical axis−similarly to single-molecule localization microscopy−paving the way for single-frame multi-plane imaging.

For the sake of completeness, other groups have already used the ISM principle in the context of STED microscopy. In particular, to improve the performance of tomographic STED microscopy[47], and to sustain higher fluorescent photon-flux[48]. However, both these works adopted sub-optimal detector arrays: the former a slow conventional camera and the latter an AiryScan-like detector. While the AiryScan detector−a hexagonally-arranged fibre bundle coupled to a series of photomultipliers (PMT) or single-element SPAD detectors−had the merit of massively spreading the ISM technique in its original computational ISM version, SPAD array detectors have everything in favour to make the definitive transition of laser-scanning microscopy, in general, to detector arrays: SPAD array detectors offer higher elements/pixels scalability, robustness, and higher photon-collection efficiency compared to those of PMTs-based AiryScan. Furthermore, in stark contrast to PMTs, they are single-photon timing detectors.

In this scenario, our STED-ISM implementation based on a SPAD array detector can become the gold standard: the proposed implementation (i) requires only minimal changes in a conventional STED microscopy architecture; (ii) preserves all functions of STED microscopy, such as multi-colour, three-dimensional, and fast imaging; (iii) is fully compatible with all current approaches for photo-damage reduction and signal-to-background ratio improvement. In terms of this last point, the single-photon timing nature of an asynchronous readout SPAD array−namely a detector with independent pixels[7]− allows the combination of STED-ISM microscopy and time-resolved STED microscopy[23,25,49] to further improve the resolution for a given STED beam intensity. Such a time-resolved STED implementation based on a SPAD array detector will provide benefits not only for imaging but also for fluorescence fluctuation spectroscopy (FFS)[50,51]: we have recently shown how the SPAD array detector improves the information content of a FFS experiment[52].

In general, this work represents a further fundamental milestone toward the transition from single-element detectors to SPAD array detectors in laser-scanning microscopy. The unique ability of a SPAD array to spatially and temporally tag fluorescence photons−from the probing volume of a laser-scanning microscope−can improve the characteristics of current advanced microscopy techniques, but open the way to novel techniques.

## Methods

### Custom setup

For this work, we updated the ISM setup described previously[11] with a STED line (Supplementary Fig. 1). The excitation beam was provided by a triggerable pulsed (~80 ps pulse-width) diode laser (LDH-D-C-640, Picoquant) emitting at 640 nm. The STED beam was provided by a femtosecond mode-locked Ti:sapphire laser (Chameleon Ultra II, Coherent) running at 775 nm. We coupled the STED laser beam into a 100 m long polarization maintaining fibre (PMF). Before injection into

the PMF, the beam passed through two 20 cm long glass rods to temporally stretch the pulse width to a few picoseconds in order to avoid unwanted nonlinear effects and damage during the fibre coupling. We used a half-wave plate (HWP) to adjust beam polarization parallel to the fast axis of the PMF. The combination of glass rods and PMF stretched the pulses of the STED beam to ≈ 200 ps. For the data shown in Fig. 6 and 12, the depletion laser is provided by a sub-nanosecond (~580 ps) fibre laser (Katana HP, Onefive GmbH) emitting at 775 nm[30]. We controlled the power of the Ti:sapphire and excitation lasers thanks to two acousto-optic modulators (AOM, MCQ80-A1,5-IR and MT80-A1-VIS, respectively, AAopto-electronic). The STED laser (master) runs at 80 MHz and provides an electronic reference signal which we used to synchronize electronically the excitation laser diode (slave). We used a picosecond electronic delayer (Picosecond Delayer, Micro Photon Devices) to temporally align the excitation pulses with respect to the depletion pulses. The STED beam emerging from the PMF was collimated, filtered in polarization by a rotating Glan-Thompson polarizing prism and phase-engineered though a polymeric mask imprinting a 0−2π helical phase-ramp (VPP-1a, RPC Photonics). We rotated a quarter-wave plate and a half-wave-plate to obtain circular polarization of the STED beam at the back-aperture of the objective lens. We co-aligned the excitation and STED beam using two dichroic mirrors (T750SPXRXT and H643LPXR, AHF Analysentechnik). After combination, the excitation and STED beams were deflected by two galvanometric scanning mirrors (6215HM40B, CT Cambridge Technology) and directed toward the objective lens (CFI Plan Apo VC 60 × , 1.4 NA, Oil, Nikon) by the same set of scan and tube lenses used in a commercial scanning microscope (Confocal C2, Nikon). The fluorescence light was collected by the same objective lens, descanned, and passed through the multi-band dichroic mirror as well as through a fluorescence band pass filter (685/70 nm, AHF Analysentechnik). A 300 mm aspheric lens (Thorlabs) focuses the fluorescence light into the pinhole plane generating a conjugated image plane with a magnification of 300×. For ISM measurements the pinhole is maintained completely open. A telescope system, built using two aspheric lenses of 100 mm and 150 mm focal length (Thorlabs), conjugates the SPAD array with the pinhole and provides an extra magnification factor. The final magnification on the SPAD array plane is 450×, thus the size of the SPAD array projected on the specimen is ~1.4 A.U. (at the far-red emission wavelength, i.e. 650 nm). Every photon detected by any of the 25 elements of the SPAD array generates a signal that is delivered through a dedicated channel (one channel for each sensitive element of the detector) to an FPGA-based data-acquisition card, which is controlled by our own software. More in detail, we controlled the microscope with the BrightEyes microscope control system (BrightEyes-MCS), a home-built LabVIEW (National Instruments, Austin, TX) program based on the Carma application (Genoa Instruments, Genoa, Italy)[30,53]. The BrightEyes-MCS (1) provides a graphical user interface to control the major acquisition parameters (e.g., scanned region, pixel size, axial position, pixel dwell time); (2) registers (in photon-counting mode) the 25 digital signals of the detector array in temporal bins of minimal 500 ns and in synchronization with the beam scanning system and other devices, e.g., laser shutters; and (3) visualizes the recorded signals (e.g., intensity images and time traces). A quite unique feature of the BrightEyes-MCS is the possibility to record, for each pixel, the fluorescence signal over multiple temporal bins. The BrightEyes-MCS stores the data in HDF5 files. We collected the data with two distinct acquisition systems. Experimental results from Figs. 3, 5, S3, S4, S5, S9, S10 are obtained with a custom SPAD array connected to a DAQ (NI USB-7856R, National Instruments)[11]. Experimental results from Figs 6, S12, S13 are obtained with the PRISM-Light kit (Genoa Instruments), which includes a detector array with microlenses−for improved collection efficiency−and the dedicated control unit. All power values reported for this setup refer to the power measured before the pair of galvanometric mirrors.

## Sample preparation

We demonstrated the enhancement in spatial resolution obtained with our STED-ISM approach on two-dimensional (2D) imaging of fluorescent beads and tubulin filaments. *Fluorescent beads*. In this study, we used a commercial sample of ATTO647N fluorescent beads with a diameter of 23 nm (Gatta-BeadsR, GattaQuant). *Tubulin filaments imaging in fixed cells*. Human HeLa cells were fixed with ice methanol, 20 min at − 20 °C and then washed three times for 15 min in PBS. After 1 hour at room temperature, the cells were treated in a solution of 3% bovine serum albumin (BSA) and 0.1% Triton in PBS (blocking buffer). The cells were then incubated with the monoclonal mouse anti-$\alpha$-tubulin antiserum (Sigma Aldrich) diluted in a blocking buffer (1:800) for 1 h at room temperature. The $\alpha$-tubulin antibody was revealed using Abberior STAR Red goat anti-mouse (1:1000, Abberior). The cells were rinsed three times in PBS for 5 min. *SiR-Tubulin in live-cells*. To label tubulin proteins, Human HeLa cell were incubated with SiR-tubululin kit (Spirochrome) diluted in LICS at a concentration of 1 $\mu$M for 30 min at 37 °C and immediately after imaged in the microscope.

## Numerical simulations

We simulated the point-spread-function of the STED-ISM system using the mathematical model presented in the Supplementary Section 2. In more detail, we used a discretized version of the image formation model using the integer indices $(n, m) \in [-2, 2]^2$ to denote the detector element. We applied the adaptive pixel-reassignment method to obtain the PSF of ISM and STED-SIM. Alternatively, we summed all scanned image PSFs to obtain the open pinhole PSF. In this case, the size of the whole array active area represents the size of the pinhole. To calculate the normalized intensity distribution of the excitation PSF, the emission PSF, and the vortex beam we used the Focus Field Calculator Matlab package[54].

Here we list all the parameters chosen for the simulations shown throughout the manuscript. The excitation, emission and depletion wavelengths are, respectively, $\lambda_{exc} = 646$ nm, $\lambda_{det} = 669$ nm and $\lambda_{STED} = 775$ nm. All PSFs are calculated in a volume of $1.27 \times 1.27 \times 1.27$ $\mu$m with $127 \times 127 \times 127$ voxels (voxel size = $10 \times 10 \times 10$ nm). We set the numerical aperture of the oil objective lens to NA = 1.4. We simulated a detector with $5 \times 5$ sensitive elements, arranged in a squared fashion, with a fill factor of 100% (we neglected any dead area between sensitive elements). We fixed the side length of the detector to 1.4 Airy units (defined by the diameter of the Airy disc, 1 AU = $1.22\lambda_{det}$/NA). For the STED PSF simulation, we used the following parameters: $\tau_F = k_F^{-1} = 3.5$ ns, $T = 1$ ns, $\varsigma = 0,10, 30,300$, where $\varsigma$ is calculated with respect to the maximum intensity value at the doughnut beam.

## Image reconstruction and analysis

To reconstruct the high-resolution STED-ISM image we used either the simple adaptive pixel-reassignment (APR) method or a multi-image deconvolution algorithm, which are fully described in Castello et al.[11]. Here, we briefly review the two methods.

The adaptive pixel-reassignment (APR) method consists of (i) shifting each scanned image from detector element $(n, m)$ by a shift vector; (ii) adding up all the shifted images. In this work, the shift vectors are directly estimated from the scanned images, without the need for any input from the user. In particular, we use a phase-correlation approach[39,55] capable of automatically taking into consideration the geometry of the detector array and the magnification of the microscope system, which can compensate for distortions (misalignments and aberrations) of the system that may arise during imaging. Very importantly in the context of this work, this automatic estimation of the shift vectors accounts for the saturation level of the STED experiment, i.e. how much the effective fluorescence volume is shrunk, without the need for laborious calibration procedures. Multi-image deconvolution is routinely used when it is necessary to fuse different microscopy images of the same sample, but characterized by different point spread functions[40,56–59]. Here, we used the multi-image generalization of the well-known Richardson-Lucy algorithm already introduced by Castello et al.[11] and further developed by Zunino et al.[60]

$$o^{k+1} = o^k \sum_{n,m} \left[ w_{n,m} \cdot h_{n,m}^* \star \left( \frac{i_{n,m}}{h_{n,m} * o^k} \right) \right] \quad (1)$$

where $\star$ is the cross-correlation operator, $*$ is the convolution operator, $h_{n,m}$ is the normalized PSF linked to the element $(n, m)$ of the SPAD array, the $*$ superscript symbol denotes the adjoint operation, $i_{n,m}$ is the scanned image generated by the same element, and $o^k$ is the reconstructed image at iteration $k$. The weight factor $w_{n,m}$ takes into account the different SNR of the scanned images and is calculated as the inverse of the fingerprint, i.e. $w_{n,m} = f_{n,m}^{-1}$. We used a simple Gaussian PSF, shifted by the quantity calculated via the phase-correlation method. The full width at half maximum (FWHM) of the PSF is fixed by the resolution value calculated by the Fourier ring correlation (FRC) algorithm applied to the STED-ISM image[41]. The same value is used as FWHM for all the PSFs. This protocol results in a sort of blind reconstruction, where no input from the user is required.

In addition, we introduced a small modification to the algorithm to take into account the term $b_{n,m}$, which is the expected background. In this case, the iterative formula is

$$o^{k+1} = o^k \sum_{n,m} \left[ w_{n,m} \cdot \overline{h}_{n,m}^* \star \left( \frac{i_{n,m}}{\overline{h}_{n,m} * o^k + b_{n,m}} \right) \right] \quad (2)$$

where $\overline{h}_{n,m}$ is the $(n, m)$ normalized PSF, but not shifted. The PSFs can be measured experimentally and shifted back using the adaptive pixel reassignment method. However, we simplified the algorithm by generating a centred Gaussian PSF with FWHM obtained using the FRC analysis. The background $b_{n,m}$ can be obtained with any of the focus-ISM implementations described in the following section. Notably, the background term can include also information about the dark noise. The expected dark noise for each element can be easily measured by registering the signal from the SPAD array detector in the absence of any source of light.

## Focus-STED algorithm

We implemented two versions of our background removal algorithm. The first version, named $f^1$-ISM, is the simplest, the fastest, but also the least accurate. It consists of evaluating pixel by pixel the background $\beta$ as the average intensity value of the external frame of the $5 \times 5$ micro-image

$$\beta(\mathbf{x}_s) = \frac{1}{16} \sum_{(n,m) \in F} i_{n,m}(\mathbf{x}_s) \quad (3)$$

where $F = \{(n, m) : |n| = 2 \vee |m| = 2\}$. Then, the in-focus signal is estimated as

$$\alpha_{n,m}(\mathbf{x}_s) = \left( \frac{1}{9} i_{n,m}(\mathbf{x}_s) - \beta(\mathbf{x}_s) \right) \cdot 25 \quad (4)$$

If, as a result, some pixels have negative values, they are trimmed to zero. The second version, named $f^2$-ISM, is more computationally demanding but is more accurate and cannot generate non-physical results. First, a region of the image containing only in-focus emitters is manually selected to calculate the in-focus fingerprint. The latter is fitted to a single Gaussian function and its standard deviation is recorded as $\sigma_{sig}$. If it is not possible to identify a region that contains only in-focus emitters, then an additional calibration measurement is needed. Alternatively, it is possible to estimate $\sigma_{det}$ theoretically. Then the adaptive pixel reassignment method is applied to the full dataset.

Subsequently, each reassigned micro-image of each pixel is normalized and fitted to the following model

$$i(\mathbf{x}_d|\mathbf{x}_s) = \underbrace{\alpha \cdot g(\mathbf{x}_d|\mathbf{0}, \sigma_{\text{sig}})}_{\text{in} - \text{focus}} + \underbrace{\beta \cdot g(\mathbf{x}_d|\mathbf{0}, \sigma_{\text{bkg}})}_{\text{out} - \text{of} - \text{focus}} \tag{5}$$

where $g(\mathbf{x}|\boldsymbol{\mu}, \sigma)$ is a normalized Gaussian function of average $\boldsymbol{\mu}$ and standard deviation $\sigma$. The weights $\alpha$ and $\beta$ follow the conservation of photon flux constraint, namely $\alpha + \beta = 1$. The standard deviation of the background micro-image $\sigma_{\text{bkg}}$ can be either selected manually or left as a free fitting parameter (see Supp. Fig. 10). In the latter case, only two parameters are free. However, individual micro-images typically have a very poor SNR. Therefore, it is good practice to restrict the minimum value of $\sigma_{\text{bkg}}$ to avoid overfitting. The parameters found with the fitting algorithm are used to build two different micro-images for each pixel: the two Gaussian functions generate, respectively, an in-focus and an out-of-focus micro-image. Eventually, the pixels of each classified micro-image are summed to generate the in-focus and out-of-focus image.

### Statistics and reproducibility

The results of our method demonstrate the capabilities to obtain high-resolution and high SNR STED-ISM images and to remove the out-of-focus light from ISM images. As such, the results do not depend on the statistical variations or the properties of the used samples. Thus, each result presented in the manuscript has been reproduced a limited number of times. More in detail

- The data of Fig. 3a and S2d has been acquired once, but the image has been acquired with a field of view large enough to contain ~ 50 beads to add statistical significance to the single-bead analysis.
- The results of Fig. 3d, e, S3, S4, S5 have been reproduced on similar samples ten times.
- The results of Fig. 5, S9, and S10 have been reproduced on similar samples five times.
- The results of Fig. 6, S12, S13, and S14 have been reproduced on the full stack of 31 planes.

All the remaining figures present simulated data that could be reproduced an indefinite number of times.

### Reporting summary

Further information on research design is available in the Nature Portfolio Reporting Summary linked to this article.

## Data availability

The experimental data generated and analysed in this study have been deposited in the Zenodo database and can be accessed at https://doi.org/10.5281/zenodo.7303679[61].

## Code availability

The code used for the current study is publicly available at the following repository https://github.com/Alessandro-Zunino/Focus-ISM[62].

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

## Acknowledgements

This research was supported by the European Research Council, BrightEyes, No. 818699 (G.T. and G.V.), and by the European Union's Horizon 2020 research and innovation program under the Marie Sklodowska-Curie grant agreement no. 794531 (AdaptiveSTED) (S.K.). We thank Prof. Alberto Tosi, Dr. Mauro Buttafava, and Federica Villa—from Politecnico di Milano—for the joint development of the SPAD array detector and for the continuous support in its optimization; Sabrina Zappone, Andrea Bucci, Marco Scotto, Dr. Mattia Donato, Dr. Eli Slenders and Dr. Eleonora Perego—from Istituto Italiano di Tecnologia—for useful discussions; Elena Tcarenkova—from University of Turku—for helpful discussions in the early stages of this project. Dr. Luca Lanzanó, Dr. Michele Oneto, and Simone Pelicci—from Istituto Italiano di Tecnologia—for support in sample preparation.

## Author contributions

G.T., A.Z., and G.V. designed the study. G.V. supervised the project with support from C.J.R.S., P.B., and A.D. G.T., F.F., S.K., A.Z., and G.V. designed and implemented the custom STED system with the SPAD array detector. G.T., and M.C. developed the control software. S.P., and M.C. integrated the SPAD array detector into the control and data-acquisition system. G.T., A.Z., M.C., S.K., and G.V., developed the analysis software. G.T., F.F., A.Z., and G.V., performed the experiments. G.T., A.Z., and G.V. analyzed the data with the support of all other authors. G.T., A.Z., and G.V. wrote the manuscript. All authors discussed the results and commented on the manuscript.

## Competing interests

M.C., S.P., P.B., A.D., and G.V. have personal financial interest (co-founders) in Genoa Instruments. The remaining authors declare no competing interests.

 
