## [Peer Review File · Nature Communications]

Focus Image Scanning Microscopy for Sharp and Gentle Super-Resolved MicroscopyREVIEWER COMMENTS

Reviewer #1 (Remarks to the Author):

The current manuscript follows up previous work on the introduction of a novel SPAD array detector and its potential in scanning fluorescence microscopy. The authors here show further potential by expanding work for efficient out-of-focus removal and super-resolution STED microscopy. I am very convinced that the combination of ISM with the novel array detector opens a lot of outstanding possibilities to microscopy with high impact, and the current manuscript greatly introduces some of them. Therefore, any further attention to this topic is important, and I therefore would be supportive of publication in Nature Communications.

The presented data is of high accuracy and very convincing – many controls and theoretical and experimental controls have been given. My biggest concern though is the wealth of information. The reader at some point loses overview of what has been done already and what is new and more importantly what is relevant to STED only and what in general. Maybe the authors could reorganize the results section of the manuscript a little, with a precise introduction of the array detector and what has been done so far with ISM, then bringing focus-ISM, as it is a general method, and then bring the benefits for STED microscopy?

Also, many results are only qualitatively shown but not quantified such as contrast improvements, resolution, SNR, ... - the case for almost all figures. Is there any chance to highlight the improvements more quantitatively than just visually?

Minor points:

- Page 2, middle: What is anti-Stokes fluorescence background?
- Introduction: Also here I got confused what is generally relevant and what specific to STED. I also for the introduction suggest first to introduce ISM and later STED.
- Page 3, bottom: Specify “small resolution enhancement” – how much exactly?
- What is meant by “asynchronous” with respect to the SPAD array – please detail?
- Page 3, top: With respect to the APR method, what has been done before and what is new?
- Begin Results section: Why should APR not be combinable with STED – any surprise that it is?
- Page 8: Briefly mention how this was calculated. This is hard to follow, since first parameters such as shift-vectors are introduced and only later explained in detail. Maybe state “as detailed below” when introducing the parameters?
- Figure 3b: Why are the shift values anisotropic – should not be from theory?
- What does ideal STED mean and what would non-ideal be?
- Page 10: Why is the first approach crude and why does it lead to negative values, and why is the second approach computationally more expensive and more precise – did not fully get this?
- Page 12: What is “conventional” out-of-focus fluorescence background?

Reviewer #2 (Remarks to the Author):

This manuscript reports about combining STED microscopy with Image Scanning Microscopy (ISM) based on a 5x5 SPAD array detector. While ISM offers a two-fold improvement of the lateral resolution (say 120 nm as a ballpark figure), STED can routinely provide a resolution of 80 nm and even 50 nm in optimal conditions. In what sense might STED profit from ISM? The drawback of STED lies in the high power of the depletion beam, required to sufficiently shrink the remaining excitation volume. This might lead to adverse effects that ultimately limit the usability and performance of STED. The use of an array detector provides two advantages to these limitations. As the authors explain, the ISM principle can enhance the resolution and by this allows to operate the depletion beam at lower power. A second advantage is achieved by using the ‘micro-image’ of the detector to discriminate between true signal (light emitted from the excited volume) and background emission coming from out-of-focus excitation and possible anti-Stokes fluorescence, excited by the depletion beam. The authors termed this method focus-ISM. Shortly, this method consists of a) finding the signal PSF

(approximated by a 2D Gaussian) that depends on the STED-saturation, and b) fitting the background contribution (a broader 2D Gaussian) to each micro-image'. Doing this for the micro-images of all scan positions, the background can be substantially reduced. This clearly helps to achieve crisper and better resolved images.

The authors explain the concept and realizations quite clearly. They demonstrate the function of the method using bead samples, and stained microtubules in fixed and live cells.

In the section 'Reducing Intensity with Adaptive Pixel-Reassignment ISM' it is nicely shown that STED-ISM using adaptive pixel-reassignment helps to reduce the power of the depletion beam for live cell imaging. However, the results are given in a rather qualitative way. It would be of great interest to potential users of this method to have a relationship between depletion power and resolution improvement for classical STED vs. STED-ISM. There probably is a sweet spot, where STED-ISM is most efficiently used.

Similarly, in the section 'Removing Background with Focus-ISM', the principles and workflow are nicely shown and possible caveats of the method are described. The benefit of the method could be shown as improvement in resolved spatial frequencies in the FRC and improvement in contrast, e.g. if S : the mean intensity of the 5% brightest pixels, and BG : the mean intensity of the 5% darkest pixels, then show $(S-BG)/BG$ for the various algorithms (STED, f1STED-ISM, f2STED-ISM, f*STED-ISM). These quantities would be of great help to assess the different methods.

In summary, I recommend the publication of the manuscript if the discussed additions are made.

Response Letter to Reviewers

We would like to thank all the reviewers for their accurate work of revision and for their useful comments and suggestions, which helped us to improve the quality of the manuscript. Please, find below a list of point-by-point answers (in **black**) to their comments (in **blue**) along with the changes performed in the revised version of the manuscript (in *italic*).

Reviewer #1:

1. My biggest concern though is the wealth of information. The reader at some point loses overview of what has been done already and what is new and more importantly what is relevant to STED only and what in general. Maybe the authors could reorganize the results section of the manuscript a little, with a precise introduction of the array detector and what has been done so far with ISM, then bringing focus-ISM, as it is a general method, and then bring the benefits for STED microscopy?

We agree with the reviewer that the results section can be improved in terms of clarity. Thus, we moved the description of the array detector and the state-of-the-art of ISM to a small introductory paragraph at the beginning of the results section, as suggested by the reviewer. Furthermore, we reshaped part of the abstract, of the introduction, and of the conclusion to state clearly that Focus-ISM is a general concept – not necessarily tied to STED microscopy. We believe that the overall clarity of the text has significantly improved.

We do not report all the changes here because of their extension. They can be found highlighted in the text.

2. Also, many results are only qualitatively shown but not quantified such as contrast improvements, resolution, SNR, ... - the case for almost all figures. Is there any chance to highlight the improvements more quantitatively than just visually?

We agree with the reviewer that the manuscript can greatly benefit from a more quantitative characterization of the presented results. To this end, we applied the Fourier Ring Correlation analysis to quantify the resolution of our images of STED-ISM. Additionally, we performed a detailed characterization of the gain in resolution versus the depletion power by measuring the FWHM of fluorescent beads (see the revised Fig. 3 and supplementary Fig. 2). We improved the study of the optical sectioning capabilities of Focus-ISM by calculating the total amount of photons on the planes of the simulated 3D PSF, both in case of ISM and STED-SIM (see revised Fig. 4). This analysis resulted in a quantitative estimation of the performance of our method, that we report as the FWHM of the optical sectioning curve. The benefits of our technique can also be inferred from the theoretical MTFs, that we show in the new Supplementary Fig. 17. Finally, we quantified the contrast improvement of the experimental images shown in the manuscript. To this end, we calculated the radial spectra of the images and demonstrated amplification of the high-frequency content (the in-focus signal) with respect to the low-frequency content. The results – quantified by calculating the integral of the spectra – follow the expected trend and are reported in the new supplementary Fig. 14.

3. Page 2, middle: What is anti-Stokes fluorescence background?

Typically, fluorescent dyes emit photons at wavelengths longer than those of the excitation beam due to non-radiative energy dissipation. However, absorption and emission *spectra* overlap: in particular, the absorption tail may not be null even at wavelengths higher than the maximum of the emission *spectrum*. It is indeed possible to excite a dye with a red-shifted excitation wavelength (e.g., 775 nm for ATTO647N), resulting in a fluorescent photon with a shorter wavelength. This phenomenon is known as anti-Stokes shift, and it is mainly caused by the excitation of vibrational levels. Although it is a highly inefficient process, its contribution becomes non-negligible if high power beams are being used, such as those of the STED depletion beam. To improve the clarity of our text we added a citation (<https://doi.org/10.1038/131839b0>) and modified the text as follows.

...thus generating additional anti-Stokes background, namely the fluorescence generated by the depletion beam [citation].

4. Introduction: Also here I got confused what is generally relevant and what specific to STED. I also for the introduction suggest first to introduce ISM and later STED.

We agree with the reviewer that discussing ISM before STED makes the reading of the introduction more linear. Thus, we changed the section accordingly. Due to the length of the modified text, we do not report it here.

5. Page 3, bottom: Specify “small resolution enhancement” – how much exactly?

While the resolution improvement of both Quantum-ISM and SOFISM techniques is theoretically unlimited, in practice the maximum gain experimentally achieved so far is about 2.5 with respect to the classical diffraction limit. To be more specific in our text, we modified the sentence as follows

However, they require a long pixel-dwell time (\geq ms) to achieve high SNR, while resolution enhancement greater than 2.5 has not been demonstrated, thus limiting their practical applications in the current development state.

6. What is meant by “asynchronous” with respect to the SPAD array – please detail?

Camera detectors are composed by an array of photosensitive pixels which are read all the same time (“global shutter”) or line-by-line sequentially (“rolling shutter”) to compose a frame. SPAD arrays could be seen as very small photon-counting cameras, but their read-out is different. Indeed, the output of each pixel of the detector can be read individually without any synchronization required with every other pixel. In other words, the read out of the SPAD array is not frame-based, enabling additional freedom in the architecture of data acquisition and in the data analysis. To improve the clarity of this detail, we added a citation (<https://doi.org/10.1364/OPTICA.391726>), moved the sentence to the conclusion section, and modified it as follows.

In terms of this last point, the single-photon timing nature of an asynchronous-readout SPAD array – namely a detector with independent pixels [citation] – allows the combination of STED-ISM microscopy and time-resolved STED microscopy...

7. Page 3, top: With respect to the APR method, what has been done before and what is new?

In brief, we previously introduced the Adaptive Pixel Reassignment method [1] to retrieve the shift-vectors directly from the recorded data. Such a strategy does not require any model of the optical system and of the photo-physical properties of the sample, which are otherwise needed to compute the expected shift vectors. In this work, we extended the use of the APR method to STED microscopy. In this context, we argued that the adaptive strategy is even more valuable since the shift-vectors show a strong dependency to the depletion power. This complication would require additional *a priori* knowledge to produce an effective predictive model to calculate the shift-vectors. To better clarify the goal of the first part of our paper, we modified our manuscript as follows.

[1]Castello, M., Tortarolo, G. et al. A robust and versatile platform for image scanning microscopy enabling super-resolution FLIM. Nat. Methods 16, 175–178 (2019).

In detail, (i) we applied the adaptive pixel-reassignment (APR) method to improve the spatial resolution while keeping a relatively low STED beam intensity, thus potentially reducing

photo-damage; (ii) we introduced a classification method to separate the in-focus from the out-of-focus background signals, thus improving optical-sectioning.

More generally, we investigated in detail the multidimensional ISM dataset from different point of views. The 25 scanned images encode the information of the shift vectors, a key element for APR reconstruction (Fig. 1c), as described in the results sections. We prove that their role is even more crucial for STED-ISM, being very sensitive to the depletion power and thus hindering different approaches of pixel reassignment. Here, we demonstrate that, after a successful reassignment, the micro-images directly encode the sample's axial position for any STED intensity. We exploited this critical insight to develop focus-ISM, a novel two-step algorithm. First, it implements the well-established APR method and then a classification algorithm to remove the out-of-focus light, while leaving intact the in-focus signal (Fig. 1d).

8. **Begin Results section: Why should APR not be combinable with STED – any surprise that it is?**

We agree with the reviewer that it is not surprising that the pixel reassignment method is compatible with STED microscopy. However, analogue implementation of PR would require complex pre-calibration steps and digital implementations using theoretical shift-vector would require a non-trivial study of the exact dependency of the shift-vectors from the depletion power. Adaptive pixel reassignment estimates the shift-vector directly from the data, inherently taking into account all the experimental conditions – among them, the depletion power. This feature makes APR the best method to combine ISM with STED. To further clarify this point, we changed the beginning of the results section as follows.

In this Section, we demonstrate how the concept of APR can be beneficial to STED microscopy, enabling a target resolution at lower STED beam intensity.

9. **Page 8: Briefly mention how this was calculated. This is hard to follow, since first parameters such as shift-vectors are introduced and only later explained in detail. Maybe state “as detailed below” when introducing the parameters?**

We agree with the reviewer that the first time the shift vectors are introduced, namely, in the comment of Fig. 1, they are not fully explained. Therefore, we followed the reviewer's suggestion of stating that the details are provided later in the text. See reply to comment n. 7 to see the changes in the text.

10. **Figure 3b: Why are the shift values anisotropic – should not be from theory?**

The reviewer is correct; the theory predicts symmetrical shift-vectors, as those depicted in Fig. 2. However, the theory and the simulations developed in this manuscript and elsewhere do not consider experimental non-idealities such as noise, optical aberrations, and misalignments. While it is necessary to minimize the presence of those non-idealities, in experimental practice they cannot be fully eliminated. The difference between expected and measured shift-vectors highlights the importance of an adaptive algorithm to perform ISM reconstruction. We highlighted this aspect by adding a citation (<https://doi.org/10.1038/s41592-018-0291-9>) and expanding the comment of figure 3b.

In general, the adaptive approach compensates for possible misalignments and other non-idealities of the optical system. Notably, the experimental shift-vectors are drastically different from the ideal ones of Fig. 2b, highlighting the crucial role of an adaptive reassignment [citation]. As discussed above, in the context of STED-ISM the APR method also accommodates implicitly the strong dependency of the shift-vectors on the STED beam powers.

11. **What does ideal STED mean and what would non-ideal be?**

With ideal STED we mean “infinite” depletion power or – in other words – point-like excitation (see also supplementary section 2.4). We recognize that this scenario cannot be experimentally achieved, but any sufficiently high depletion power can be approximated to this model, enabling a great simplification of the explanation of the working principle of focus-ISM. In practice, any realistic STED experiment is non-ideal. The feasibility of focus-ISM for non-ideal STED and conventional ISM is explained later in the manuscript.

In this Section, we regard at the ISM dataset from the perspective of the micro-images in order to effectively improve the optical sectioning of ISM. Our new method, named focus-ISM (fISM), removes the out-of-focus background analyzing each micro-image. First, we show the working principle and the feasibility of fISM considering the case of ideal STED, namely the case of point-like excitation (or, equivalently, infinite depletion power). Later, we generalize our method to STED microscopy at any depletion power and to conventional ISM. As a matter of fact, our method is well suited also to any ISM-based technique.

12. Page 10: Why is the first approach crude and why does it lead to negative values, and why is the second approach computationally more expensive and more precise – did not fully get this?

The first approach essentially approximates the out-of-focus fingerprint as flat, with a constant value given by the average of the outer pixels of the detector. This model is not physical, and we cannot expect it can correctly estimate the background content of each micro-image. If a fraction of the signal is detected by the outer pixels, then the background is overestimated, and the subtraction can generate negative values. On the other hand, the background removal procedure is a simple difference (see also the Methods section) and can be quickly calculated by any modern computer. The second approach enforces a more physical model of the out-of-focus fingerprint, namely a broad Gaussian distribution. In addition, the algorithm enforces the conservation of energy, preventing the generation of pixels with negative values. However, the second approach requires applying a fitting procedure to each micro-image. The fitting is an iterative procedure that typically requires a computational time much higher than that of the first approach. To improve the clarity of our manuscript, we modified it as follows.

The out-of-focus light is distributed broadly across the detector elements, and outer elements do not register in-focus light. This observation suggests a simple way to estimate the background -- namely using a flat out-of-focus micro-image. Our first algorithm (f¹-ISM) estimates the background by calculating the average of the signal collected by the outer elements of the detector array. The background value is then subtracted from the inner elements. In detail, we use all the outer-ring elements to extract the average background per scan-point. The simplicity behind this strategy, which only requires a few basic arithmetic operations, enables a fast estimation of the background, paving the way for real-time focus-ISM. However, the approximation behind this approach is crude and might lead to non-physical results, such as pixels with negative photon counts.

13. Page 12: What is “conventional” out-of-focus fluorescence background?

With conventional background, we mean out-of-focus fluorescence light. Since we already specify in the text that we are discussing out-of-focus light, we removed the word “conventional” which we find unnecessary. To improve the clarity of our text, we modified the manuscript as follows.

So far, we have discussed only the out-of-focus fluorescence background generated by the excitation of the sample at positions different from the focal plane. However, the depletion beam can generate non-negligible anti-Stokes fluorescence background possibly originating from any axial plane.

Reviewer #2:

1. In the section 'Reducing Intensity with Adaptive Pixel-Reassignment ISM' it is nicely shown that STED-ISM using adaptive pixel-reassignment helps to reduce the power of the depletion beam for live cell imaging. However, the results are given in a rather qualitative way. It would be of great interest to potential users of this method to have a relationship between depletion power and resolution improvement for classical STED vs. STED-ISM. There probably is a sweet spot, where STED-ISM is most efficiently used.

We agree with the reviewer that the results were originally reported too qualitatively. Therefore, we added an FRC analysis of the resolution of our images, now reported on the images in Fig. 3. Furthermore, we studied the resolution and signal enhancement of STED-SIM with respect to STED as a function of the depletion power (see the revised Fig. 3, the supplementary Fig. 2, and the reply to the 2nd comment of the 1st reviewer). It is apparent from the graphs that the benefits of pixel reassignment – in terms of resolution and signal – monotonically decrease with the depletion power, such that a sweet spot cannot be clearly identified. However, pixel reassignment is a necessary step to apply Focus-ISM and thus improve the optical sectioning of the imaging technique, making the combination of STED with ISM extremely useful in terms of enhancing the signal-to-background ratio.

2. Similarly, in the section 'Removing Background with Focus-ISM', the principles and workflow are nicely shown and possible caveats of the method are described. The benefit of the method could be shown as improvement in resolved spatial frequencies in the FRC and improvement in contrast, e.g. if S: the mean intensity of the 5% brightest pixels, and BG: the mean intensity of the 5% darkest pixels, then show $(S-BG)/BG$ for the various algorithms (STED, f1STED-ISM, f2STED-ISM, f*STED-ISM). These quantities would be of great help to assess the different methods.

We agree with the reviewer that the optical sectioning enhancement required more detailed analysis. Thus, we added to the manuscript a detailed study of such improvement by calculating the optical sectioning curves and their width in the case of ISM and STED-ISM (see the revised Fig. 4). Furthermore, we simulated the MTFs of our system and compared it to the MTF resulting from the application of Focus-ISM (see the new supplementary Fig. 17). Finally, we provided a tailored metric to quantify the contrast enhancement of our experimental results. Unfortunately, the metric suggested by the reviewer would not be useful for our results. Indeed, it is not uncommon that the out-of-focus background is brighter than the in-focus signal (e.g., see Fig. 5). Thus, a metric based on the pixel intensity value cannot be used as an estimator of the contrast enhancement in our images. Instead, we demonstrated the amplification of the high-frequency content (the in-focus signal) with respect to the low-frequency content by calculating the integral of the normalized spectra of the images. The results are reported in the new supplementary Fig. 14. See also the reply to the 2nd comment of the 1st reviewer for further information.

REVIEWER COMMENTS

Reviewer #1 (Remarks to the Author):

The authors have well replied to all of my previous comments and revised the manuscript accordingly. It is an important story which now reads very well. I suggest publication as is.

Reviewer #2 (Remarks to the Author):

In the revised version of the manuscript my main concerns and questions have been satisfactorily addressed. I can now recommend the manuscript for publication.